# Bacterial-archaeal co-occurrence in honey bee gut microbiomes across host species and management regimes

**Jian-Ping Ying**[ID]°, **Yang-Feng Zou**°, **Tao Jiang***, **Jia-Li Chang**[ID]*

Department of New Energy Materials and Chemistry, Leshan Normal University, Leshan, China

° Contributed equally to this work.
* jialiyangfeng@163.com (J-LC); composting@163.com (TJ)

## Abstract

Honeybees are key pollinators, and their overwintering period represents a critical bottleneck for colony survival. We investigated how host species and husbandry practices influence the composition of gut bacterial and archaeal communities of overwintering honeybees, as well as the potential functional consequences for energy metabolism and resilience. Using 16S rRNA amplicon sequencing combined with marker-gene functional inference, we observed that wild *Apis cerana* (wAc) harbors the most diverse gut microbiome across both bacterial and archaeal domains. Notably, wAc exhibited a significant enrichment of methanogenic archaea (e.g., *Methanocorpusculaceae* and *Methanosarcinaceae*), a pattern potentially consistent with bacterial–archaeal metabolic coupling that may improve fermentation thermodynamics and energy extraction under winter resource limitation. In contrast, the gut communities of managed *Apis cerana* (mAc) and *Apis mellifera* (Am) were dominated by *Lactobacillus*, and mAc exhibited a relative increase in predicted carbohydrate metabolism and replication/repair pathways based on marker-gene inference. Most archaeal sequences from Am and mAc remained unclassified, underscoring gaps in primer coverage and reference databases. Because each experimental group in this study was represented by a single pooled sample, the analyses are descriptive and hypothesis-generating rather than definitive; functional inferences should be treated as provisional and validated in future work. Overall, the results generate testable hypotheses that dietary diversification, reduced antibiotic exposure, and targeted microbial interventions might help support overwintering resilience, but targeted validation is required before making management recommendations.

## 1 Introduction

Honeybees are among the most widespread organisms in terrestrial ecosystems and human food systems [1]. As primary pollinators of a wide range of fruits, vegetables,

**Data availability statement:** The sequencing data generated in this study have been deposited in the China National Center for Bioinformation (CNCB-NGDC) under BioProject accession PRJCA048519. The associated BioSample accessions are SAMC5999114–SAMC5999116 (and the archaeal community samples SAMC5999117 and SAMC5999119). Data can be accessed at the CNCB-NGDC BioProject page: https://ngdc.cncb.ac.cn/bioproject/browse/PRJCA048519.

**Funding:** National Natural Science Foundation of China (No. 52200056).

**Competing interests:** The authors have declared that no competing interests exist.

and oilseed crops, they support both wild-plant reproduction and agricultural productivity. Global estimates indicate that the majority of the world's crop species benefit from animal-mediated pollination, with honeybees providing the dominant share of managed pollination services [2]. Consequently, two bee taxa have become focal points of both applied and basic research: the western honeybee (*Apis mellifera*, hereafter Am) and the eastern honeybee complex (principally *Apis cerana* and its closely related lineages, hereafter Ac). Although both groups perform the essential ecological function of pollination, they differ markedly in their life histories, physiologies, and the manner in which humans manage them. *A. mellifera* has been widely transported and intensively managed on a global scale for commercial pollination [3], with large colony sizes, frequent translocations, monocultural foraging landscapes, and recurrent chemical interventions (e.g., acaricides and antibiotics) characterizing many Am operations [4–6]. In contrast, *A. cerana* commonly persists as wild, locally adapted populations or as regionally managed stocks maintained under diverse husbandry regimes. Ac populations forage in heterogeneous landscapes and are exposed to distinct diseases and climatic conditions [7]. These biological and anthropogenic contrasts create unique host–environment–microbiome contexts that may influence physiology, disease susceptibility, and resilience to seasonal stressors.

Overwintering is one of the most critical phases in the annual cycle of honeybee colonies, and represents a pronounced bottleneck for population persistence [8]. As temperatures decline and floral resources become scarce, colonies must strike a delicate balance between conserving energy and maintaining the internal conditions necessary for brood survival and worker activity [9]. Behavioral thermoregulation—most notably the formation of compact overwintering clusters and the generation of metabolic heat—enables colonies to maintain a warm core even when ambient temperatures fall below 10 °C [9]. These strategies, however, are energetically expensive and require substantial autumnal accumulation of nectar- and pollen-derived reserves; colonies that fail to secure adequate stores or that experience disruptions in foraging or health are at an elevated risk of overwinter loss [10]. Common beekeeper interventions (e.g., supplemental feeding, insulation, and Varroa control) can mitigate some risks, but they may also alter nutritional inputs and microbial exposures [6]. Notably, interspecific differences are important; some eastern honeybee populations sustain foraging and activity at lower temperatures and exhibit physiological traits that confer greater cold tolerance than many western stocks [11,12]. These differences may interact with host-associated microbiota to influence overwintering outcomes.

The gut microbiome is increasingly recognized as a central regulator of insect health and a mediator of ecological and physiological plasticity [12]. In honeybees, a relatively low-diversity but functionally specialized bacterial core—dominated by taxa such as *Snodgrassella*, *Gilliamella*, *Lactobacillus* spp., and *Bifidobacterium* spp.—is involved in pollen-wall degradation, carbohydrate fermentation, vitamin synthesis, and colonization resistance against pathogens [13–16]. These functions can directly influence host nutrition and immune status during periods of nutritional stress such as winter. For example, fermentative bacteria generate short-chain fatty acids [17] and other metabolites [18,19] that can be utilized by the host or by other microbial

partners, whereas certain core taxa competitively exclude opportunistic pathogens that proliferate when the host defenses are compromised [20]. In addition to bacteria, archaea, particularly methanogenic lineages, are emerging as potentially important yet understudied partners. By consuming hydrogen and formate produced during bacterial fermentation, methanogens can enhance thermodynamically challenging fermentation pathways and improve fermentation efficiency through mutually beneficial interactions [21]. Such interdomain metabolic syntrophy could be especially important under overwintering conditions, where efficient extraction of energy from limited stores is critical. Management and diet profoundly influence the gut microbiome, and by extension, host physiology. Managed colonies often receive sugar syrup or pollen substitutes, experience higher colony densities, and are exposed to chemical treatments that can perturb microbial communities. These practices can select opportunistic or fast-growing taxa (e.g., certain lactobacilli) while diminishing overall diversity and functional redundancy [22]. In contrast, wild colonies typically forage on a more diverse suite of floral resources and encounter a broader environmental microbiome, which can promote microbial diversity and retain the eco-functional capacity [23]. As microbiome composition mediates nutrient processing, detoxification, and immune modulation, structural differences in microbiome between wild and managed bees—particularly during overwintering—may translate into measurable differences in energy balance, pathogen resistance, and ultimately survival.

Motivated by these ecological and management-relevant considerations, the present study compared the gut bacterial and archaeal communities of overwintering *A. mellifera* (Am), wild *A. cerana* (wAc), and managed *A. cerana* (mAc). Using 16S rRNA amplicon sequencing, we quantified alpha- and beta-diversity, characterized taxonomic composition across multiple ranks, and applied marker-based functional prediction (Kyoto Encyclopedia of Genes and Genomes [KEGG] classifications) to infer potential metabolic capacities associated with each host–management combination. Our objectives were to determine (i) whether host species and management status predictably structure microbial diversity during overwintering, (ii) which bacterial and archaeal taxa distinguish wild from managed eastern and western bees, and (iii) which inferred functional pathways may plausibly support overwinter metabolic demands. By integrating taxonomic and functional perspectives, we aimed to identify candidate microbial processes that could contribute to overwinter resilience and inform microbiome-aware strategies for supporting colony health, including targeted dietary interventions and microbiome restoration approaches.

## 2 Materials and methods

### 2.1 Sample collection and preservation

At the time this study was conducted, there were no specific national or institutional regulations in China governing the care and use of insects such as honey bees, and formal approval from an animal ethics committee was therefore not required. All bees were collected on January 2, 2021, from managed and wild colonies in Daying County, Sichuan Province, China. Access to apiary and field sites was obtained with the prior permission of the colony owners and was approved by Leshan Normal University. According to current national and local regulations, no additional government permits are required for non-destructive sampling of non-protected invertebrate species outside protected areas; this study did not involve endangered or protected species or protected habitats.

For the sequencing analysis we collected ten adult worker bees per group. For Am (*Carniolan* — a non-native strain originating from the Alps and Balkan Peninsula, procured from the Jilin Province Institute of Apicultural Science, Jilin, China), five colonies were randomly chosen from a pool of 200 colonies after excluding diseased or weak colonies, and two adult workers were randomly sampled from each selected colony (n = 10). For mAc (locally established, artificially captured and bred bees), five colonies were randomly selected from 30 available colonies after excluding diseased or weak colonies, and two adult workers were randomly sampled from each colony (n = 10). For wAc (wild, locally native bees), because of the limited number of available colonies and the practical difficulty of field collection, only three colonies were located in areas isolated from managed Western and captive Eastern honeybee populations; a total of ten adult worker

bees were collected across these wild colonies (n = 10). Specimens from each group were placed in labeled centrifuge tubes and maintained at −20 °C during transport to the laboratory.

## 2.2 Extraction of sample DNA

The intestinal contents collection method was slightly modified based on the previous report [24]. Collected bees were assigned to three groups: Am, mAc, and wAc. After surface sterilization with 75% ethanol, individual bees were transferred to sterile Petri dishes inside a laminar-flow hood. Using sterile forceps, the intestines were dissected and placed into centrifuge tubes containing phosphate buffer. The intestinal wall was punctured with a sterile syringe and the tubes were agitated for 15 minutes to release the gut contents. The gut wall was then removed with a sterile needle, and the resulting bacterial suspension was used for genomic DNA extraction.

For sequencing, intestinal contents from ten bees were pooled to create a single composite sample per group (Am, n = 1; mAc, n = 1; wAc, n = 1). DNA was extracted following the manufacturer's protocol for the FastDNA™ Spin Kit for Feces. Pooling reduces individual-to-individual variation and therefore facilitates between-group comparisons, but it also prevents estimation of within-group variability; as a result, calculation of confidence intervals and $P$-values for individual-level effects is not possible. Nonetheless, this approach provides useful comparative insight into gut microbiota differences among species and management practices during overwintering.

## 2.3 PCR and sequencing for 16S rRNA

The V3-V4 hypervariable region of the bacterial 16S rRNA gene was amplified using primer pair 338F (5′-ACTCCTAC GGGAGGCAGCAG-3') and 806R (5′-GGACTACHVGGGTWTCTAAT-3') [25]. The V4 region of the 16S rRNA gene of archaea was amplified using primers 524F (5'-TGYCAGCCGCCGCGGAA-3') and 958R (5'-YCCGGCGTTGAVTC CAATT-3') on an ABI GeneAmp 9700 PCR thermal cycler (ABI, USA) [25]. The PCR reaction mixture included: 4 µL 5 × Fast Pfu buffer, 2 µL 2.5 mM dNTPs, 0.8 µL of each primer (5 µM concentration), 0.4 µL Fast Pfu polymerase, 10 ng template DNA, and ddH$_2$O to a final volume of 20 µL. PCR amplification cycle conditions were as follows: initial denaturation at 95°C for 5 minutes, followed by 34 cycles, each cycle including denaturation at 94°C for 30 seconds, annealing at 57°C for 30 seconds, extension at 72°C for 60 seconds, and final extension at 72°C for 10 minutes. All samples were amplified in triplicate independently. PCR products were purified using the AxyPrep DNA Gel Extraction Kit (Axygen Biosciences, USA) and quantified using the Quantus Fluorometer (Promega, USA). Sequencing libraries were generated under quality control using the TruSeq DNA PCR-Free Sample Preparation Kit (Illumina, USA). Finally, the purified amplification fragments were mixed in equimolar ratios. Paired-end sequencing (2 × 300 bp) was performed on the Illumina MiSeq platform at Shanghai Ouyi Biomedical Technology Co., Ltd. (Shanghai, China) according to standard procedures.

## 2.4 Data processing and bioinformatics analysis

Raw FASTQ files were first filtered to remove reads containing ambiguous bases (N) and reads with single-nucleotide repeat (homopolymer) runs ≥ 10 bp. Paired-end reads were then merged using FLASH (v1.2.11). Merging parameters were set as minimum overlap = 15 bp and maximum mismatch ratio in the overlap = 0.1 (i.e., 10%); merged tags shorter than 15 bp or exceeding the allowed mismatch threshold were discarded. After merging, sequences were quality-controlled: reads were filtered with max expected errors (maxEE) = 1.0, bases with quality score < 20 were truncated, reads containing any ambiguous base (N) were removed, and sequences containing homopolymer runs ≥ 10 bp were discarded. Cleaned tags were clustered into operational taxonomic units (OTUs) (ASV methods (e.g., DADA2/Deblur) work best with multiple biological replicates; because each group here has only one pooled sample, ASV adds little for assessing within-group variation or replication-based inference, so we used a conservative OTU workflow) using VSEARCH (v2.17.1) [26] at 97% sequence identity. Chimera removal was performed using the UCHIME [27] algorithm implemented

in VSEARCH (sensitive settings). Representative OTU sequences were taxonomically annotated with the RDP Classifier (v2.2) against the SILVA SSU 138.1 reference database, using a confidence threshold of ≥ 0.6. OTU-based Venn diagrams, abundance accumulation curves, and species accumulation curves were plotted in R. α-diversity indices (Shannon, Simpson, Chao, coverage) were computed using mothur and plotted in R. β-diversity (clustering and PCA) analyses were performed in QIIME 2 (v2021.8) [28]. To control for library size differences, data used for descriptive visualization were normalized to within-sample relative abundances (sum-scaling to percentages). Because each experimental group consisted of a single pooled sample (n = 1), no rarefaction or replicate-based statistical standardization was performed. Finally, the PICRUSt tool was used for functional inference, which predicts function based on taxonomic annotations of 16S rRNA from the Kyoto Encyclopedia of Genes and Genomes (KEGG) and COG databases.

## 3 Results

### 3.1 Data summary and operational taxonomic unit (OTU) distribution

After sequencing, reads were trimmed, filtered, and subjected to quality control. The three bacterial datasets yielded a total of 2,282 OTUs, with effective sequence ratios of 91.82%, 89.06%, and 86.43% for Am, wAc, and mAc samples, respectively. Archaeal sequencing produced 545 OTUs. Effective sequence ratios were 93.23%, 91.71%, and 95.16% for Am, wAc, and mAc samples, respectively. Bacterial abundance was markedly higher than archaeal abundance across all three samples, confirming that bacteria constitute the core microbial group in the intestines of western honeybees and Chinese honeybees. The OTU count for the Am sample was 827, significantly lower than that of wAc (1,380) and mAc (1,574). The distribution of bacterial and archaeal OTUs across groups is shown in Fig 1. The three gut bacterial communities shared 417 OTUs, representing the core microbiota (Fig 1A). Among them, the Western honeybee (Am) had 231 unique OTUs, the Chinese honeybee (wAc) has 413 unique OTUs, and mAc had 556 unique OTUs. Pairwise overlaps were also evident: Am–wAc shared 64 OTUs, Am–mAc shared 115 OTUs, and wAc–mAc shared 486 OTUs. In contrast, Fig 1B shows that the overall archaeal diversity was comparatively low, with only 24 core OTUs shared among the 3

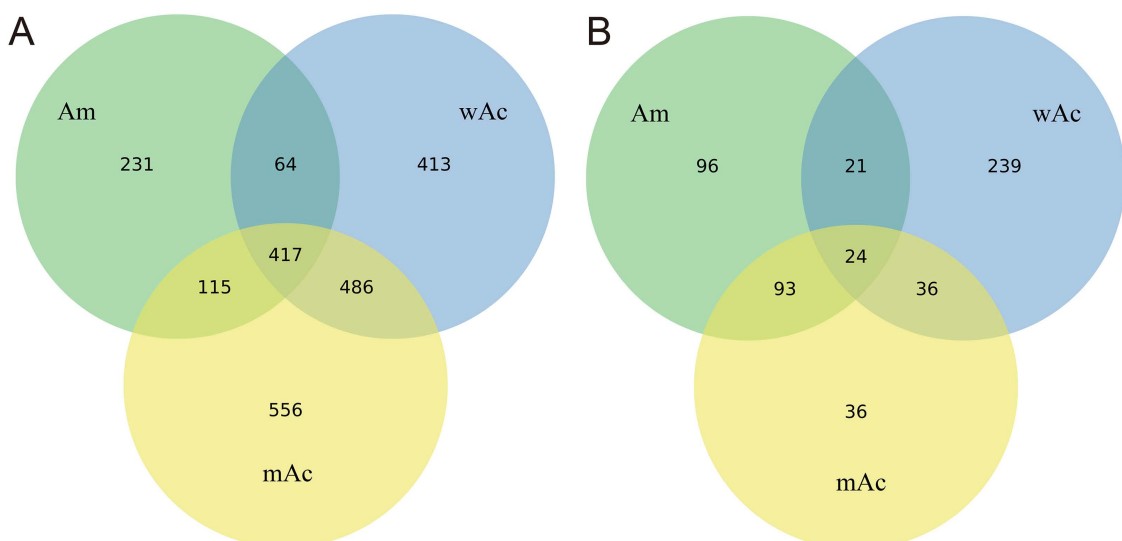

**Fig 1. Venn diagram of OTU counts derived from 16S sequencing of honey bee guts from different groups.** A: Shared and unique bacterial OTU counts; B: Shared and unique archaeal OTU counts. The values in the figure represent the number of shared and unique OTUs between groups (sample group: Am = *A. mellifera*, wAc = wild *A. cerana*, mAc = managed *A. cerana*). Each group is represented by a single pooled sample (n = 1); consequently, these results are descriptive and do not permit statistical inference of within-group variation.

groups. Unique archaeal OTUs included 96 in Am, 239 in wAc, and 36 in mAc, while pairwise overlaps were 21 for Am–wAc, 93 for Am–mAc, and 36 for wAc–mAc.

### 3.2 Composition of gut microbiota in Am, wAc, and mAc during the wintering period

Across samples, the intestinal microbiome was numerically dominated by a small number of high-abundance taxa, although the bacterial and archaeal fractions exhibited distinct compositional patterns among groups.

At the phylum level (Fig 2A), bacterial communities were primarily represented by *Firmicutes* and *Proteobacteria*, with markedly different contributions across groups. In mAc, *Firmicutes* comprised 60.3% of the sequence reads, followed by *Bacteroidetes* (17.2%) and *Proteobacteria* (15.8%) as the main secondary phyla. In contrast, Am and wAc displayed a more even phylum composition: Am contained *Firmicutes* (44.6%), *Proteobacteria* (42.9%), and *Actinobacteria* (10.6%), whereas wAc contained *Firmicutes* (44.7%), *Proteobacteria* (24.2%), *Bacteroidetes* (15.4%), and *Actinobacteria* (15.3%).

The patterns at the class (S1A Fig) and order (S1B Fig) levels reflected these differences. Bacilli were the most abundant class across the groups (Am, 43.7%; mAc, 52.9%; and wAc, 39.0%). Proteobacterial representation was mainly driven by Alphaproteobacteria and Gammaproteobacteria (e.g., Alphaproteobacteria, 38.8% in Am; Gammaproteobacteria, 22.4% in wAc). At the order level, Lactobacillales dominated each group (Am, 43.7%; mAc, 52.9%; wAc, 39.0%), whereas orders affiliated with Proteobacteria and Bacteroidetes (e.g., Rhizobiales and Bacteroidales) contributed substantially to Am and wAc but were comparatively depleted in mAc.

Family- (S1C Fig) and genus-level (Fig 2B) analyses revealed pronounced enrichment of lactobacilli. Lactobacillaceae (family) and *Lactobacillus* (genus) were the largest taxonomic components in all groups, with family-level abundances of 43.7%, 52.8%, and 38.9% for Am, mAc, and wAc, respectively, and corresponding genus-level abundances of *Lactobacillus* (Am,43.7%; mAc, 52.8%; wAc, 39.0%). Secondary taxa varied by group: Rhizobiaceae (38.6%) were unusually abundant in Am, whereas wAc contained notable proportions of Orbaceae (20.4%) and Bifidobacteriaceae (14.9%). At the genus level, in addition to *Lactobacillus*, *Gilliamella,* and *Bifidobacterium* were detectable as secondary contributors, especially in wAc.

Species-level assignments (S1D Fig) for bacteria were constrained by ambiguous or uncultured annotations; for example, in mAc, "Other" and "Ambiguous_taxa" accounted for 53.8% and 20.1%, respectively, whereas in wAc, "Ambiguous_taxa" and "Other" accounted for 40.0% and 21.5%. Among the resolved species, *Lactobacillus* spp. SF6D was relatively abundant in wAc (18.8%) and to a lesser extent in Am (10.5%), whereas uncultured *Lactobacillus*-type entries accounted for a notable proportion of Am (28.4%). These results indicate pervasive Firmicutes dominance largely driven by *Lactobacillus*, with the Firmicutes/*Lactobacillus* enrichment most pronounced in mAc, and larger Proteobacterial and Actinobacterial contributions in Am and wAc, respectively.

The archaeal composition differed markedly among the groups and exhibited variable annotation resolution. In Am, the archaeal fraction was predominantly composed of broadly unclassified assignments at the phylum level (Fig 2C), with "Other" accounting for 81.2%, *Euryarchaeota* 14.3% and *Proteobacteria* 3.5%. At the class level (see S2A Fig), annotations were dominated by "Other" (82.8%) and Methanomicrobia (12.3%). Order-level annotations (S2B Fig) were also nonspecific, with the majority of reads assigned to the "Other" (81.2%) followed by Euryarchaeota (14.2%). Family- (S2C Fig) and genus-level (Fig 2D) annotations were dominated by aggregated labels (family "Other" 83.1%; genus "g_Other" 84.9%), whereas recognized methanogenic families such as Methanosarcinaceae (6.4%) and Methanocorpusculaceae (5.9%) represented minor fractions. The species-level assignments (S2D Fig) in Am were largely unresolved (e.g., "Ambiguous_taxa," 8.0%), indicating that most archaeal diversity in Am remains unclassified at finer taxonomic ranks under the current reference mapping.

In mAc, the archaeal component was almost entirely composed of high-level ambiguous annotations. At the phylum level (Fig 2C), "Other" accounted for 98.6%, with only trace assignments to *Proteobacteria* (1.5%) and *Euryarchaeota*

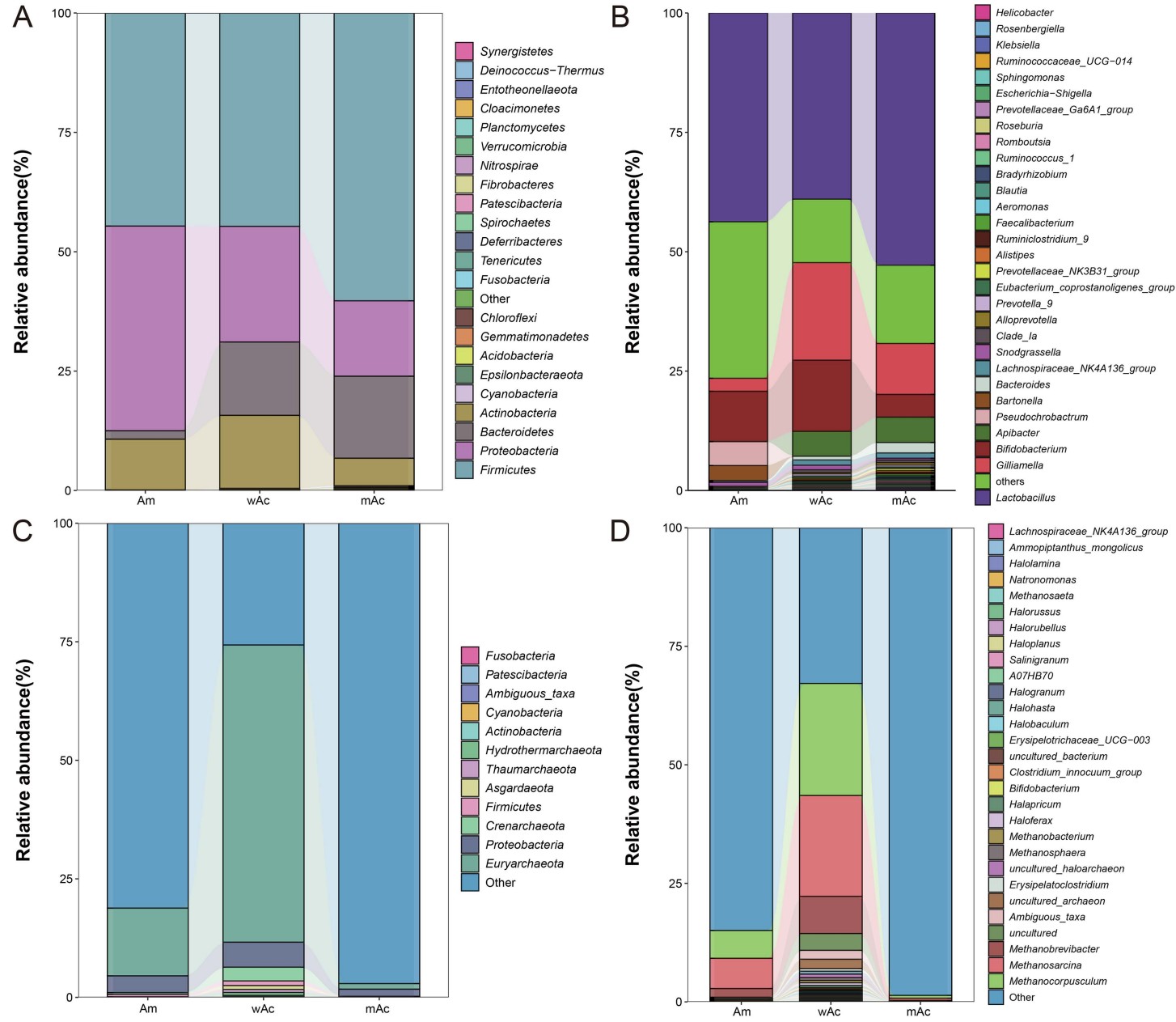

**Fig 2. Composition of the honey bee gut microbiome (relative abundance, %) from different groups.** A: Bacterial phylum composition (stacked bar chart showing major phyla and "Other"); B: Bacterial genus composition (showing the most abundant genera, with the rest grouped as "Other"); C: Archaea composition; D: Archaea composition. The legend indicates the phylum/genus name corresponding to each color, and the labels indicate relative abundance percentages. Relative abundance (%) based on total effective sequence readings. Each group is represented by a single pooled sample (n = 1); consequently, these results are descriptive and do not permit statistical inference of within-group variation.

(1.2%). Class- (S2A Fig), order- (S2B Fig), family- (S2C Fig), and genus-level (Fig 2D) annotations were dominated by "Other" or "g_Other" labels (≈98% at each rank), and species-level reads (S2D Fig) were similarly unresolved ("Other," 98.6%). Under the current annotation pipeline and reference set, taxonomic inference for mAc at the order, family, genus, and species levels is therefore not robust.

In contrast, wAc exhibited a clear archaeal signature that was consistent across taxonomic ranks. At the phylum level (Fig 2C), *Euryarchaeota* accounted for 62.7% of reads, followed by "Other" (25.7%) and *Proteobacteria* (5.3%). Order-level assignments (S2B Fig) in wAc were more informative than those in Am or mAc, reflecting methanogen-dominated classes and families. Methanomicrobia comprised 45.3% of class-level reads (S2A Fig) while family-level profiles (S2C Fig) were dominated by Methanocorpusculaceae (23.6%) and Methanosarcinaceae (21.3%) and their corresponding genera (*Methanocorpusculum* and *Methanosarcina*) as shown in Fig 2D. However, at the species-level (S2D Fig), resolution in wAc remained limited: "Other" (34.0%), "Ambiguous_taxa" (31.1%) and an "uncultured_archaeon" cluster (30.2%) together accounted for the majority of species-level reads, indicating that a substantial portion of archaeal diversity remains uncharacterized at the species rank.

Collectively, archaeal composition ranged from nearly entirely unresolved in mAc to a methanogen-enriched profile in wAc, with Am dominated by unclassified lineages and only a modest methanogenic signal. A recurrent limitation across all groups is the high fraction of reads annotated as "Other," "Ambiguous_taxa" or uncultured at order, genus, and species levels, which constrains fine-scale ecological interpretation.

### 3.3 Discrepancies in the gut microbiota of Am, wAc, and mAc during the wintering period

Alpha diversity metrics revealed clear differences in richness and diversity among the groups for both bacteria and archaea (Table 1). Good's coverage was 0.99 for all groups, indicating that sequencing depth was sufficient to capture the vast majority of community members. For bacteria, richness (Chao1) increased sequentially from Am (1,460.22) to wAc (1,572.38) to mAc (1,778.78). Shannon diversity was the highest in wAc (4.81), slightly lower in mAc (4.59), and the lowest in Am (3.50). The Simpson index mirrored the Shannon index, indicating greater evenness in wAc (Simpson = 0.89) compared to mAc (0.77) or Am (0.83). Together, these results show that wAc harbors the most diverse and even bacterial community, whereas Am exhibits lower diversity and evenness despite its intermediate richness, consistent with dominance by a few abundant taxa. The archaeal diversity showed an even more pronounced group effect. Chao1 richness was highest in wAc (324.57), intermediate in Am (243.67), and lowest in mAc (212.14). Shannon diversity and Simpson evenness were also highest in wAc (Shannon = 4.91; Simpson = 0.88) but markedly lower in Am (Shannon = 3.06; Simpson = 0.61) and especially mAc (Shannon = 1.68; Simpson = 0.33). Thus, the archaeal community in wAc was both richer and more evenly distributed, whereas that in mAc was impoverished and dominated by a few taxa. In summary, wAc consistently exhibited the highest alpha diversity in both domains; mAc showed the highest bacterial richness but reduced evenness; and Am displayed relatively low bacterial and intermediate archaeal diversity.

Principal-coordinate analysis (only used to visually illustrate the overall compositional differences among the three groups; not intended for statistical inference) of community composition revealed clear group-specific structuring for both bacteria (S3A Fig) and archaea (S3B Fig). For bacteria (S3A Fig), the first two axes accounted for the vast majority of variance (PC1 = 65.6%, PC2 = 34.4%). Samples from Am were separated from wAc and mAc primarily along PC1, whereas

**Table 1. Analysis of α diversity of intestinal microbiota in different groups based on 16S rRNA gene sequences\*.**

|          | Sample ID | Chao1    | Goods_coverage | Shannon | Simpson |
|----------|-----------|----------|----------------|---------|---------|
| Bacteria | Am        | 1,460.22 | 0.99           | 3.50    | 0.83    |
|          | wAc       | 1,572.38 | 0.99           | 4.81    | 0.89    |
|          | mAc       | 1,778.78 | 0.99           | 4.59    | 0.77    |
| Archaea  | Am        | 243.67   | 0.99           | 3.06    | 0.61    |
|          | wAc       | 324.57   | 0.99           | 4.91    | 0.88    |
|          | mAc       | 212.14   | 0.99           | 1.68    | 0.33    |

\* All α-diversity values represent point estimates from a single sample and thus do not include uncertainty or error margins.

mAc was further distinguished from wAc along PC2 — together indicating that Am harbors a bacterial assemblage compositionally distinct from the other two groups, and that wAc and mAc differ in a second orthogonal compositional axis. For archaea (S3B Fig), ordination also showed strong separation (PC1 = 71.5%, PC2 = 28.6%): wAc occupies the negative extreme of PC1, while mAc and Am lie on the positive side of PC1 and are separated from one another along PC2 (mAc positive, Am negative). Collectively, these ordinations indicate that both bacterial and archaeal communities differ by group, with between-group differences captured by the first one or two principal coordinates.

### 3.4  Functional differences in Am, wAc, and mAc gut flora during the wintering period

The predicted KEGG functional profiles at level 2 (L2) revealed consistent, group-specific differences in the inferred metabolic potential (Fig 3). The top-20 L2 pathways for each group are shown in Fig 3, with the full L2 results provided in S4 Fig. In bacteria (Fig 3A), membrane transport, carbohydrate metabolism, replication and repair, translation, and amino acid metabolism were the dominant categories across groups, but their relative abundances differed: Am was enriched in membrane transport functions, mAc showed relatively higher representation of carbohydrate metabolism and replication/repair pathways, and wAc displayed intermediate position between the two. At the archaeal level (Fig 3B), the translation and amino acid metabolism remained the dominant categories, whereas methanogen-associated categories (e.g., cofactor/vitamin and energy metabolism) were enriched in the group with the strongest methanogenic taxonomic signals.

Predicted COG functional profiles similarly revealed a shared core of housekeeping and metabolic functions, accompanied by biologically meaningful shifts among groups (Fig 4). In bacteria (Fig 4A), the dominant COG categories included translation, ribosomal structure, and biogenesis (J); carbohydrate transport and metabolism (G); amino acid transport and metabolism (E); replication, recombination, and repair (L); and energy production and conversion (C). They together constituted a majority of the predicted assignments. Despite this conserved core, Am exhibited relatively higher proportions of transport-related and broadly annotated functions (R and S), mAc showed greater contributions to carbohydrate metabolism and replication/repair, and wAc was intermediate. Archaeal COG profiles (Fig 4B) mirrored these

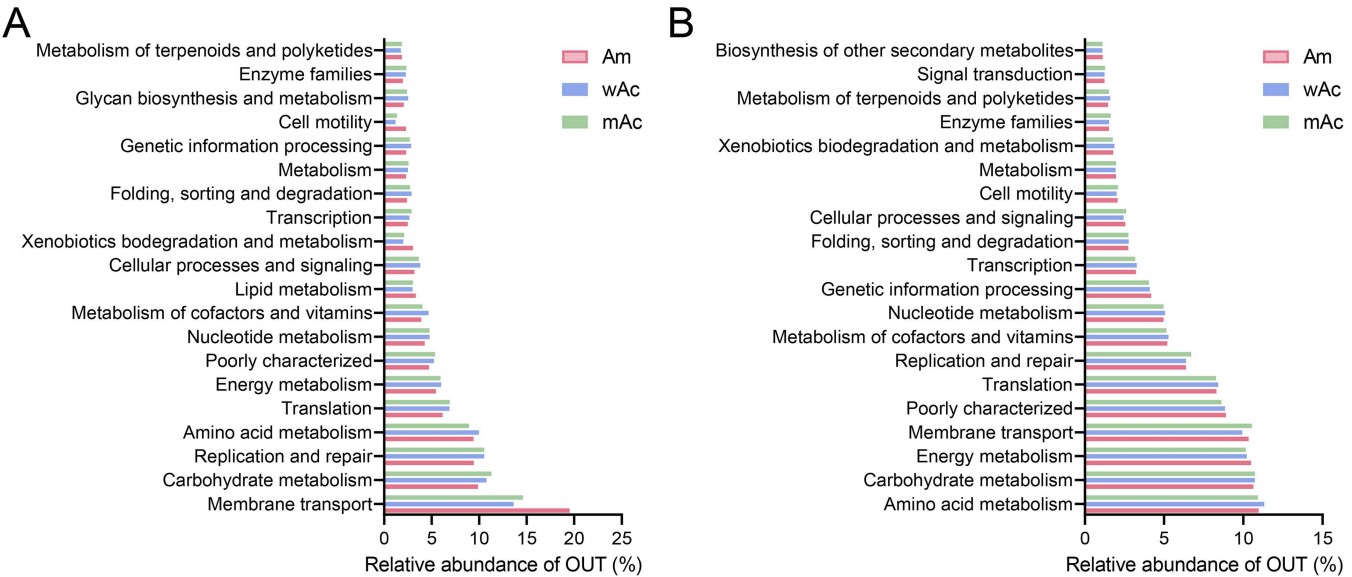

**Fig 3.  KEGG functional annotation of the honey bee gut microbiome from different groups (Level 2, based on 16S marker gene prediction).**
A: Bacterial functional annotation (showing the relative abundance of the top 20 L2 pathways); B: Archaeal functional annotation (showing significantly enriched or biologically relevant L2 categories). Each group is represented by a single pooled sample (n = 1); consequently, these results are descriptive and do not permit statistical inference of within-group variation.

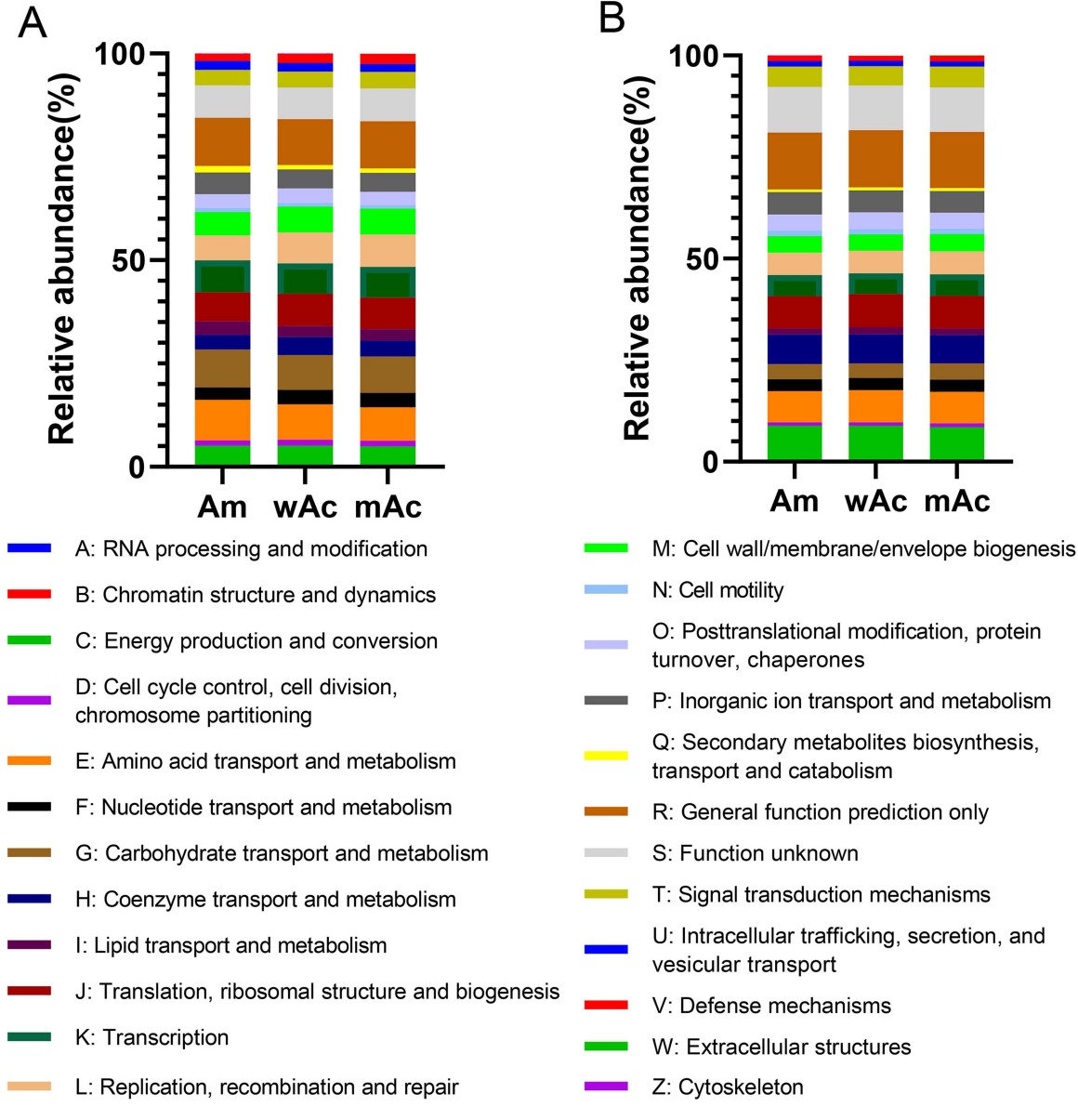

**Fig 4. COG functional annotation of the gut microbiome of different honey bee groups.** A: Bacterial COG category distribution (showing the main COG classes and their relative abundance); B: Archaeal COG category distribution (highlighting energy metabolism, cofactor-related, and unannotated functions). Each group is represented by a single pooled sample (n = 1); consequently, these results are descriptive and do not permit statistical inference of within-group variation.

patterns: translation (J), energy production and conversion (C), and amino acid metabolism (E) dominated overall, while wAc, the methanogen-enriched group, exhibited a relatively higher representation of energy- and cofactor-related COGs (C and H). In contrast, mAc and Am displayed larger proportions of general function predictions (R) and unknown functions (S), reflecting the prevalence of unclassified archaeal sequences in these groups.

Both KEGG Level-2 (L2) and COG functional predictions indicate a conserved core of shared functions across the groups, accompanied by group-specific shifts—most notably in transport systems, DNA replication/repair, and

energy-related pathways—that are consistent with the taxonomic differences described above. However, a substantial fraction of predicted annotations in both Am and mAc were classified as "other/unclear," and marker-gene-based inference is constrained by reference-database coverage and annotation quality. Therefore, these pathway-level observations should be interpreted with caution and require validation using higher-resolution approaches (e.g., shotgun metagenomics, metatranscriptomics, or targeted functional assays).

## 4 Discussion

Our comparative analysis of the overwintering gut microbiome in western honey bees (*Apis mellifera*, Am), wild eastern honey bees (*A. cerana*, wAc), and managed eastern honey bees (*A. cerana*, mAc) revealed a strong influence of host species and management practices on microbial community structure and function. These findings provide insights into the potential microbial underpinnings of overwintering resilience, a critical determinant for colony survival.

The dominant members of the honeybee gut microbiome identified in this study—*Bifidobacterium (Actinobacteria), Lactobacillaceae, Gilliamella (Gammaproteobacteria), Bartonella and Bombella (Alphaproteobacteria), and Snodgrassella (Betaproteobacteria)*—were consistent with recent surveys of core bee-associated taxa [29] indicating that host species and husbandry practices do not erase the canonical gut consortium. Nonetheless, management substantially altered community abundance. mAc exhibited a pronounced enrichment of *Lactobacillus*, plausibly reflecting anthropogenic drivers such as artificial feeding with sucrose/syrup or pollen substitutes [30], high-density husbandry [31] and routine chemical interventions [32]. These practices provide abundant simple carbon sources and intermittent inputs of exogenous microbes, favoring fast-growing lactic acid bacteria specialized for rapid sugar metabolism [33]. This dynamic led to the rapid expansion of a few dominant taxa, accompanied by reduced functional diversity and redundancy. Functional inference supports this interpretation: mAc communities were relatively enriched in carbohydrate metabolism and replication/repair pathways, consistent with community optimization for the rapid exploitation of simple substrates. In contrast, although *Lactobacillus* remained a major component of wAc (~38.9%), its secondary community was more compositionally heterogeneous, featuring *Gilliamella*, *Bifidobacterium*, and Orbaceae, along with elevated proportions of *Proteobacteria* and *Actinobacteria*. This broader composition corresponded with higher Shannon and Simpson indices than observed in mAc. Such balanced diversity likely preserves functional redundancy and network stability, which are especially valuable during overwintering when nutritional inputs are scarce and metabolic resilience is required to maintain basal physiology and resist opportunistic pathogens [34]. An anomalous observation in this dataset was the unusually high abundance of Rhizobiaceae (38.6%) in Am, a pattern uncommon in canonical bee cores. This enrichment may reflect site-specific environmental exposure, different floral sources, or anthropogenic contamination (e.g., the introduction of soil- or plant-associated microbes through foraging or feed). Collectively, these results suggest that managed colonies converge toward a narrower, highly efficient metabolic configuration. This may be advantageous under certain husbandry regimes but expensive in terms of ecological and physiological plasticity.

The most striking and novel finding of this study was the pronounced divergence of the archaeal communities among the three groups. While the archaeal profiles of Am and mAc were dominated by unclassified or ambiguously annotated sequences, the wAc archaeal assemblage was markedly richer and more compositionally distinct, with clear enrichment of methanogenic lineages, particularly members of *Methanocorpusculaceae* and *Methanosarcinaceae*. This pattern strongly suggests bacterial–archaeal syntrophy in wAc, which appeared absent or disrupted in managed colonies Methanogens consume hydrogen and formate generated by bacterial fermentation, thereby altering the thermodynamics of fermentative pathways, and enhancing the efficiency of energy extraction from ingested substrates [35]. Under overwintering conditions, when energy stores are constrained and metabolic demands increase, such interdomain metabolic coupling could provide a meaningful adaptive advantage [36] and help to explain the superior cold tolerance and overwinter survival frequently observed in wild populations. However, the high proportion of unannotated archaeal reads in Am and mAc highlights a critical methodological limitation: commonly used primers and reference databases (e.g., SILVA (*v*138.1)) may

incompletely represent honeybee-associated archaeal diversity. Consequently, the apparent paucity of classified archaea in managed bees may either reflect genuine loss or annotation blind spots. Future work should combine archaeal-targeted amplification and quantitative PCR with shotgun metagenomics, long-read sequencing, and functional assays (e.g., methane flux measurements or stable isotope probing) to improve both taxonomic resolution and metabolic validation.

The bacterial and archaeal communities in wAc consistently exhibited the highest alpha diversity, evenness, and patterns with clear ecological and functional implications. High microbial diversity and functional redundancy are hallmarks of resilient ecosystems. When bee colonies face energy limitations and relative immunosuppression, a diverse gut microbiome is better able to maintain a steady flux of essential metabolites (e.g., short-chain fatty acids [37] and vitamins [38]), buffer perturbations through overlapping biochemical capabilities, and resist opportunistic pathogens via competitive exclusion. These mechanistic contributions provide a parsimonious microbiological explanation for the remarkable robustness of wild *A. cerana* populations. In contrast, the relatively low diversity of Am may reflect cumulative effects of intensive management and selective breeding [30]. Global translocation, standardized feeding, husbandry, and selection of production-oriented traits can constrain local microbial assemblages, potentially eroding the microbial variation required for rapid adaptation to regional environmental stressors [39]. In other words, management-driven standardization may inadvertently trade ecological plasticity for short-term husbandry gains.

This study provides a strong empirical basis for shifting beekeeping practices toward a more microbiome-central approach. Beekeepers could explore strategies that promote gut microbiome diversity and stability rather than relying solely on chemical interventions and simple sugar feeding—for example, providing a diverse natural diet, using microbiome transplantation for environmental adaptation, and reducing antibiotic use. The findings from wAc suggest that archaea play an integral role in the gut, providing a new area for honeybee research. Future studies should prioritize functional validation through methanogenesis experiments and stable isotope probing to confirm the role of archaea in host energy metabolism. By integrating these diverse approaches, research can advance from descriptive microbial surveys to mechanistic understanding of how gut microbiota support honeybee health and resilience, ultimately providing a more comprehensive and sustainable beekeeping framework.

## 5  Conclusion

This study provides a comprehensive overview of the gut microbiome in overwintering honeybees and highlights how host species and management practices shape these communities. Wild eastern honeybees (*Apis cerana*, wAc) consistently exhibited the most diverse and balanced bacterial and archaeal profiles. Notably, wAc was enriched in methanogenic archaea, which work in cooperation with bacteria to maximize energy extraction from food reserves, an essential adaptation for winter survival. In contrast, the gut communities of managed eastern honeybees (mAc) and western honeybees (*Apis mellifera*, Am) were dominated by *Lactobacillus*, reflecting adaptation to sugar-based artificial diets. Although these microbes efficiently metabolize simple carbohydrates, reduced diversity may compromise the resilience of bees to stress and pathogens. The findings highlight the critical role of a diverse and stable gut microbiome in maintaining honeybee health during overwintering, though intensive management practices may compromise this microbial support system. Understanding differences between wild and managed microbiomes can inform microbiome-friendly beekeeping strategies, such as dietary diversification and minimizing chemical interferences. Strengthening the gut microbiome may enhance survival and sustain the ecological contributions of honeybees.

## Supporting information

**S1 Fig. Composition of gut bacterial communities across honey bee groups.** Relative abundance (%) of bacterial taxa in the three groups (Am = *Apis mellifera*; wAc = wild *A. cerana*; mAc = managed *A. cerana*). Panels show taxonomic distributions at four ranks: (A) class, (B) order, (C) family, and (D) species. Bars represent the mean relative abundance for each group, and low-abundance taxa are pooled as "Other." Relative abundance (%) based on total effective sequence

readings. Each group is represented by a single pooled sample (n = 1); consequently, these results are descriptive and do not permit statistical inference of within-group variation.
(PDF)

**S2 Fig. Composition of gut archaeal communities across honey bee groups.** Relative abundance (%) of archaeal taxa in the three groups (Am = A. mellifera; wAc = wild A. cerana; mAc = managed A. cerana). Panels show taxonomic distributions at four ranks: (A) class, (B) order, (C) family, and (D) species. Bars represent the mean relative abundance for each group, and low-abundance taxa are pooled as "Other." Relative abundance (%) based on total effective sequence readings. Each group is represented by a single pooled sample (n = 1); consequently, these results are descriptive and do not permit statistical inference of within-group variation.
(PDF)

**S3 Fig. Principal-coordinate analysis of gut microbial community composition.** Principal-coordinate analysis (PCA) plots of 16S rRNA gene-based community composition for (A) Bacteria and (B) Archaea. Each point represents one pooled sample per group (Am = *A. mellifera*; wAc = wild *A. cerana*; mAc = managed *A. cerana*), and spatial separation among points reflects compositional dissimilarity among groups. Only used to visually illustrate the overall compositional differences among the three groups; not intended for statistical inference.
(PDF)

**S4 Fig. Heatmap of predicted functional profiles inferred from marker-gene data and annotated using KEGG at Level 2.** Rows correspond to KEGG Level 2 functional categories, and columns correspond to the three sample groups. Color intensity indicates the normalized relative abundance of each predicted functional category.
(PDF)

## Author contributions

**Conceptualization:** Tao Jiang, Jia-Li Chang.

**Data curation:** Jian-Ping Ying, Yang-Feng Zou.

**Investigation:** Jian-Ping Ying, Yang-Feng Zou.

**Methodology:** Jian-Ping Ying, Tao Jiang, Jia-Li Chang.

**Resources:** Jia-Li Chang.

**Software:** Jian-Ping Ying, Yang-Feng Zou.

**Writing – original draft:** Jian-Ping Ying, Yang-Feng Zou, Tao Jiang, Jia-Li Chang.

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
