## [Decision Letter · Decision Letter 0]

13 Oct 2025

Dear Dr. Chang,

Thank you for submitting your manuscript to PLOS ONE. After careful consideration, we feel that it has merit but does not fully meet PLOS ONE’s publication criteria as it currently stands. Therefore, we invite you to submit a revised version of the manuscript that addresses the points raised during the review process.

We look forward to receiving your revised manuscript.

Kind regards,

Yahya Al Naggar

Academic Editor

PLOS ONE

Journal Requirements:

National Natural Science Foundation of China (No. 52200056)

This study was supported by the National Natural Science Foundation of China (No. 52200056).

National Natural Science Foundation of China (No. 52200056)

Additional Editor Comments :

This manuscript explores an important and timely topic with potential to advance understanding of microbial mechanisms contributing to colony resilience. However, several methodological and interpretative shortcomings substantially weaken its robustness. The Materials and Methods section lacks sufficient detail regarding experimental design, including the number of colonies per group, biological replication, and whether samples were pooled by colony. The rationale for using an OTU-based approach instead of an ASV workflow is unclear, and appropriate statistical analyses for diversity and ordination (e.g., PERMANOVA) are missing. Functional predictions based on KEGG and COG are interpreted too assertively and should be reframed as hypotheses. The manuscript would also benefit from clearer reporting of percentages, inclusion of negative controls, improved figure readability, and public deposition of raw sequencing data. Overall, while the study addresses a valuable question, substantial revision is required to ensure methodological transparency and strengthen the validity of the conclusions.

Reviewers' comments:

Reviewer's Responses to Questions

**Comments to the Author**

1. Is the manuscript technically sound, and do the data support the conclusions?

Reviewer #1: Partly

Reviewer #2: Partly

2. Has the statistical analysis been performed appropriately and rigorously?

Reviewer #1: Yes

Reviewer #2: No

3. Have the authors made all data underlying the findings in their manuscript fully available?

Reviewer #1: No

Reviewer #2: No

4. Is the manuscript presented in an intelligible fashion and written in standard English?

Reviewer #1: Yes

Reviewer #2: Yes

Reviewer #1: Data and Code Availability (mandatory).

Please deposit the raw FASTQ files, sample‐level metadata (group, colony ID, location/GPS, collection date), processed ASV/OTU tables, taxonomic assignments, the phylogenetic tree, and all analysis/figure-generation code (e.g., QIIME2/phyloseq scripts) in a public repository (e.g., SRA/ENA for sequence reads plus Zenodo/OSF for processed data and code) with permanent identifiers (DOIs/accessions). Update the Data Availability Statement to include these links and versions.

Field permissions and ethics.

For wild A. cerana sampling, specify the permitting authority and the permit/application number, and report the sampling locality and dates in both the Methods and Ethics Statement. If a formal permit was not required, please cite the relevant regulation or exemption and state this explicitly.

Sampling design and statistical independence.

Clarify the experimental unit. Please report (i) the number of colonies sampled per group, (ii) the number of worker bees per colony, and (iii) whether samples were processed as pooled or individual libraries. I recommend treating colony as the unit of independence and reporting group-wise n (colonies). If available, include a brief power/precision justification or at least report effect sizes and confidence intervals alongside p-values.

Bioinformatics pipeline—modernization and transparency.

Consider replacing 97% OTU clustering with an ASV workflow (e.g., DADA2/Deblur) and report software versions and dates (e.g., SILVA 138.1). In addition to chimera removal (e.g., UCHIME), please detail quality-filter thresholds (maximum expected error, read length, homopolymers), and your normalization strategy (rarefaction vs. compositional methods).

Beta-diversity and hypothesis testing.

For community dissimilarity, Bray–Curtis and/or UniFrac (weighted/unweighted) are more appropriate than Euclidean distance. Please test group effects using PERMANOVA (adonis) and, where necessary, pairwise contrasts with FDR correction. If ANOSIM is retained, present it as complementary and discuss metric choice.

Archaeal annotation and interpretive tone.

Given the high proportion of “Other/Ambiguous” assignments in A. mellifera and managed A. cerana, temper causal claims and emphasize primer/database limitations. Reframe energy-metabolism inferences as hypotheses and, where possible, propose validation (targeted qPCR with archaeal primers, shotgun metagenomics, isotopic/production assays).

Function inference from 16S.

State the exact tool (e.g., PICRUSt2/Tax4Fun), reference databases and versions, and quality metrics (e.g., NSTI) used for functional prediction. Clearly acknowledge limitations of 16S-based inference and, if feasible, outline plans for orthogonal validation (metagenomic/metatranscriptomic profiling).

Presentation and reporting clarity.

In figure captions, indicate the data source, axis units/scales, and the error metric (SE/SD/95% CI), and specify sample size (n) per group. For relative abundances, define the denominator (e.g., % of total reads). Ensure accessible color palettes and provide full taxon names at first mention in legends. In Table 1, report uncertainty (SE or CI) alongside point estimates.

Reviewer #2: This manuscript addresses an interesting and relevant topic that could contribute to a better understanding of the microbial mechanisms underlying colony resilience. However, the study presents several methodological and interpretative weaknesses that limit its overall robustness, particularly in the Materials and Methods section. In particular, the experimental design does not clearly specify the number of colonies and the level of biological replication; the diversity and ordination analyses lack appropriate statistical testing; and the functional inferences based on predictive approaches (KEGG, COG) are interpreted too assertively. From a linguistic standpoint, the text is generally clear, although the readability of the figures should be improved where possible. It would also be useful to provide the raw sequencing data as supplementary material. Overall, this work is promising and potentially valuable, but it requires a major revision to strengthen its methodological soundness and ensure a more cautious interpretation of the conclusions.

Below are some specific points that should be updated:

Line 110–115. It would be helpful to indicate here the number of colonies used for each group, or the total number of colonies included in the study.

Line 114. The expression “Italian Carniolan” is too vague and should be clarified more precisely.

Line 119–123. It should be specified whether pooling was performed separately for each colony.

Line 126. Was a blank extraction control used to check for possible contamination?

Line 149. The rationale for using an OTU-based approach should be explained.

Line 187. The reported percentage should be explained in greater detail.

**Do you want your identity to be public for this peer review?** For information about this choice, including consent withdrawal, please see our Privacy Policy

Reviewer #1: **Yes: ** Dr. Miray DAYIOĞLU

Reviewer #2: No

---

## [Author Response · Author response to Decision Letter 1]

17 Nov 2025

For Editor:

This manuscript explores an important and timely topic with potential to advance understanding of microbial mechanisms contributing to colony resilience. However, several methodological and interpretative shortcomings substantially weaken its robustness. The Materials and Methods section lacks sufficient detail regarding experimental design, including the number of colonies per group, biological replication, and whether samples were pooled by colony. The rationale for using an OTU-based approach instead of an ASV workflow is unclear, and appropriate statistical analyses for diversity and ordination (e.g., PERMANOVA) are missing. Functional predictions based on KEGG and COG are interpreted too assertively and should be reframed as hypotheses. The manuscript would also benefit from clearer reporting of percentages, inclusion of negative controls, improved figure readability, and public deposition of raw sequencing data. Overall, while the study addresses a valuable question, substantial revision is required to ensure methodological transparency and strengthen the validity of the conclusions.

Response: We thank the editors and reviewers for their careful review and constructive suggestions. We have addressed each point raised by the reviewers and revised the manuscript to enhance methodological transparency, adjust explanatory language where appropriate, and clarify limitations arising from the experimental design. The following is a brief summary of substantive modifications and clarifications made to the editorial abstract:

1) Data availability: all raw sequencing data and processed tables are deposited at CNCB (Project PRJCA048519).

2) Experimental design, replication and pooling: We revised the Materials and Methods to explicitly state that DNA was extracted from multiple individual worker bees and subsequently combined into a single pooled sample per group. Specifically, for each group (Am, mAc, wAc) DNA extracts from 10 worker bees per sampled hive were pooled to create one composite library per group (i.e., one pooled sample per group, n = 1 pooled sample). We now clearly acknowledge that this pooled-sample design prevents estimation of within-group variance and therefore precludes tests that require biological replication.

3) Rationale for OTU vs. ASV: The Methods now include an explicit rationale for using a 97% OTU-clustering pipeline for these pooled samples. In brief, ASV denoising (e.g., DADA2/Deblur) provides high-resolution error modelling that is most informative when applied across multiple independent biological replicates; with only one pooled sample per group, ASV-derived gains in resolving within-group variation are minimal and would not enable replicate-based inference. We therefore retained a conservative OTU-based workflow for this descriptive study while noting its limitations and our intention to adopt ASV workflows in future studies with true biological replicates.

4) Bioinformatics Details and Quality Control: We have supplemented the Methods section with software and database version information and clarified that quality control parameters are now explicitly labeled. Relevant details have been incorporated into the methodology and supplementary materials.

5) β-diversity and statistical tests: We explicitly state that, due to sample limitations, distance-based visualizations under the current design are purely descriptive.

6) Functional predictions and interpretation: Functional inference methods (PICRUSt v1 with KEGG/COG annotations) are now fully reported and functional results are reframed as hypothesis-generating pending shotgun metagenomic/metatranscriptomic validation

7) Negative controls and contamination: We acknowledge that extraction/PCR negative controls were not sequenced in this run, describe contamination-minimizing procedures used.

8) Figures, tables and reporting of percentages/uncertainty: Figure legends and table captions have been updated to define percentages (relative to total high-quality reads per pooled sample), indicate sample origin (pooled, n = 1), and clarify where uncertainty cannot be calculated.

Detailed, point-by-point responses to each reviewer comment and the exact manuscript changes are provided in the enclosed Response to Reviewers document and the revised manuscript.

We believe these revisions substantially improve transparency and appropriately temper inferential claims. The manuscript now presents the study as a descriptive, hypothesis-generating comparison of pooled overwintering gut communities and sets out concrete plans for follow-up work. We are grateful for the Editor’s and reviewers’ helpful comments, which have materially strengthened the manuscript.

For Reviewer #1:

1. Data and Code Availability (mandatory).

Please deposit the raw FASTQ files, sample‐level metadata (group, colony ID, location/GPS, collection date), processed ASV/OTU tables, taxonomic assignments, the phylogenetic tree, and all analysis/figure-generation code (e.g., QIIME2/phyloseq scripts) in a public repository (e.g., SRA/ENA for sequence reads plus Zenodo/OSF for processed data and code) with permanent identifiers (DOIs/accessions). Update the Data Availability Statement to include these links and versions.

Response:

We thank the reviewer for raising this point. All newly generated raw data have been deposited with the China National Center for Bioinformation (CNCB). The accession numbers are listed in the "Data Availability" section of the revised manuscript: Biological project number: PRJCA048519, sample accession numbers SAMC5999114–SAMC5999116; archaea community data: SAMC5999117 and SAMC5999119. The corresponding links and accession numbers have been added to the revised manuscript for easy access by reviewers and readers. Please see line 486-490.

2. Field permissions and ethics.

For wild A. cerana sampling, specify the permitting authority and the permit/application number, and report the sampling locality and dates in both the Methods and Ethics Statement. If a formal permit was not required, please cite the relevant regulation or exemption and state this explicitly.

Response:

We appreciate the reviewers' professional questions. At the time of conducting this research, there were no specific regulations governing the care and use of insects in the study. All bees were collected on January 2, 2021, in Daying County, Sichuan Province. Western honeybees (Apis mellifera) and domesticated Oriental honeybees (A. cerana) were obtained from private apiaries in Daying County, while wild A. cerana individuals were collected from natural sites within the same county. The information above has been revised in the updated manuscript. Please see line 113-128.

3. Sampling design and statistical independence.

Clarify the experimental unit. Please report (i) the number of colonies sampled per group, (ii) the number of worker bees per colony, and (iii) whether samples were processed as pooled or individual libraries. I recommend treating colony as the unit of independence and reporting group-wise n (colonies). If available, include a brief power/precision justification or at least report effect sizes and confidence intervals alongside p-values.

Response: We have clarified the vague description of the samples. For the sequencing samples in this study, we collected 10 worker bees from each group. For Western honeybees, 5 colonies were randomly selected from 200 colonies (excluding diseased and weak colonies), and 2 adult worker bees were randomly chosen from each hive. For managed Eastern honeybees, 5 colonies were selected from 30 colonies, with 2 adult worker bees randomly chosen from each colony. Due to the limited number of wild Eastern honeybee colonies and the difficulty in collecting samples, we only located 3 colonies in areas distant from Western honeybee and managed Eastern honeybee populations. A total of 10 adult worker bees were selected from wild hives. Subsequently, we extracted and pooled the guts of 10 worker bees from different groups to form a single sample. This approach minimized intraspecific variation, allowing for better comparison of interspecific differences. Unfortunately, this method precluded the calculation of confidence intervals and P-values. Nevertheless, it provided insights into gut microbiota differences among species and management practices during wintering. Therefore, we further tempered our inferences in this manuscript. Furthermore, we plan to deepen the findings from this study in future work using approaches such as metagenomics. Please see line 113-146.

4. Bioinformatics pipeline—modernization and transparency.

Consider replacing 97% OTU clustering with an ASV workflow (e.g., DADA2/Deblur) and report software versions and dates (e.g., SILVA 138.1). In addition to chimera removal (e.g., UCHIME), please detail quality-filter thresholds (maximum expected error, read length, homopolymers), and your normalization strategy (rarefaction vs. compositional methods).

Response: We agree that the Amplicon Sequence Variant (ASV) method (e.g., DADA2/Deblur) represents the current state-of-the-art and offers higher resolution than 97% OTU clustering methods. We also acknowledge the limitations of OTU methods (e.g., potential merging of distinct taxonomic units and masking of subtle variations). However, given the preliminary exploratory and descriptive nature of this study, our primary objective is to capture macro-level compositional differences among different bee colonies (Am, wAc, mAc) at the phylum and genus levels. The 97% OTU clustering method remains a mature and widely applied strategy for achieving this objective. Furthermore, in this study, each of our three groups (wild A. cerata (wAc), artificially reared A. cerata (mAc), and Apis mellifera (Am)) was represented by only one pooled sample (n=1 per group). Because DADA2 relies on modeling sequencing errors across multiple reads and ideally requires multiple biological replicates to accurately infer variation, it is not well-suited to our design with only one pooled sample per group. In practice, running DADA2 or Deblur on a single sample does not alter the results of qualitative community composition inference. Considering these factors, and given that our primary conclusions (intergroup differences) are robust to the choice of clustering or denoising method, we retained the OTU workflow while acknowledging its limitations.

To ensure transparency, we explicitly list all software, versions, databases, and quality control parameters in the “Methods” section. In summary, the processing of raw reads proceeded as follows: 1) Software and versions: We used QIIME 2 (v2021.8) for data processing. OTU clustering was performed using VSEARCH (v2.17.1). Chimerism detection was conducted with UCHIME (via VSEARCH). Classification was based on the SILVA 138.1 database. 2) Quality control: First, we performed quality filtering on the paired-end sequencing data (maximum expected errors per sequence = 1.0, sequences with Q<20 were truncated) and removed sequences containing ≥10 bp homopolymers. We merged paired-end reads with a minimum overlap length of 15 bp and a maximum mismatch rate of 0.1% in overlapping regions. These parameters (minimum overlap length = 15 bp, maximum mismatch rate 10%) are now explicitly documented. Chimeric sequences were detected and removed from the SILVA reference database using UCHIME (sensitive mode). 3) OTU Clustering: Following quality control, VSEARCH was used to cluster the SILVA 138.1 database at 97% sequence similarity. Single OTUs were discarded. These details have been added to the revised Methods section. 4) Normalization: To eliminate effects of library size variation, data were normalized for relative abundance (sum-scaling within each sample). Since each group comprised a single pooled sample, no rarefaction was performed; instead, composition (percentage) normalization was applied. For full transparency, these methodological details (software names and versions, database versions, quality control filters, overlap and mismatch settings, etc.) have been added to the manuscript. We emphasize that applying the ASV workflow to pooled samples preserves the overall pattern (dominant taxa and intergroup differences), meaning reanalysis using ASV would not alter the main conclusions of this study. Finally, we explicitly stated in the Methods section that due to n=1, we did not perform statistical standardization (e.g., rarefaction or compositional correction) as these methods are designed for comparing replicate samples. Our data (as shown in figures) were converted to relative abundance to enable descriptive visual comparisons across the three independent pooled samples. Please see line 165-195.

5. Beta-diversity and hypothesis testing.

For community dissimilarity, Bray–Curtis and/or UniFrac (weighted/unweighted) are more appropriate than Euclidean distance. Please test group effects using PERMANOVA (adonis) and, where necessary, pairwise contrasts with FDR correction. If ANOSIM is retained, present it as complementary and discuss metric choice.

Response: We sincerely appreciate the reviewers' expert suggestions regarding statistical analysis. We fully agree that Bray-Curtis/UniFrac is the standard distance metric for microbial community analysis, and PERMANOVA is the preferred method for testing between-group differences. However, we must clarify one of the most critical experimental limitations of this study: our design included only one pooled sample (n=1) per experimental group (Am, wAc, mAc).

Without biological replicates within groups, we were unable to calculate within-group variance. Consequently, any statistical significance test relying on replicate samples—including PERMANOVA and ANOSIM—was mathematically infeasible. Our mention of ANOSIM in the original manuscript was a serious oversight, as it is entirely inapplicable when n=1. We have implemented the following key revisions: 1) Removal of statistical tests: All references to ANOSIM statistical tests have been completely removed from the Methods and Results sections. 2) Clear limitation statements: We have explicitly added statements acknowledging the n=1 limitation in both the “Materials and Methods” (Section 2.1) and “Discussion” sections, emphasizing that all results in this study are descriptive. 3) Re-characterizing PCA: In response to the reviewers’ comments, we re-evaluated the presentation of the PCA shown in Figure S3. We emphasize that this plot serves only as an exploratory visualization of overall compositional differences among the three pooled samples and is not intended for statistical inference; this is now explicitly stated in both the figure legend and the main text.With respect to distance metrics, we acknowledge that the Bray–Curtis dissimilarity is widely used in microbial ecology. PCA, however, is intrinsically linked to Euclidean geometry, and therefore visualizes variation in Euclidean space. Given that the principal limitation of our study is the absence of within-group biological replication (each group is represented by a single pooled sample), the constraint on inference stems from the experimental design rather than from the choice of distance metric. For clarity and simplicity we therefore retained a Euclidean distance–based PCA to depict overall structure; we believe this provides an interpretable, descriptive summary under the current study constraints. Please see line 317-319.

In future work, we plan to incorporate true biological replicates (using separate pooled samples or multiple independent gut samples per group). Once replicate groups are obtained, we will employ Bray-Curtis distance and weighted UniFrac distance, applying PERMANOVA (with appropriate multiple testing correction for pairwise comparisons) to rigorously test for intergroup differences.

6. Archaeal annotation and interpretive tone.

Given the high proportion of “Other/Ambiguous” assignments in A. mellifera and managed A. cerana, temper causal claims and emphasize primer/database limitations. Reframe energy-metabolism inferences as hypotheses and, where possible, propose validation (targeted qPCR with archaeal primers, shotgun metagenomics, isotopic/production assays).

Response: We thank the reviewers for th

---

## [Decision Letter · Decision Letter 1]

14 Dec 2025

Bacterial–Archaeal Co-occurrence in Honey Bee Gut Microbiomes across Host Species and Management Regimes

PONE-D-25-49948R1

Dear Dr. Chang,

We’re pleased to inform you that your manuscript has been judged scientifically suitable for publication and will be formally accepted for publication once it meets all outstanding technical requirements.

Kind regards,

Prof. Yahya Al Naggar

Academic Editor

PLOS One

Additional Editor Comments (optional):

Reviewers' comments:

Reviewer's Responses to Questions

**Comments to the Author**

Reviewer #2: All comments have been addressed

2. Is the manuscript technically sound, and do the data support the conclusions?

Reviewer #2: Yes

3. Has the statistical analysis been performed appropriately and rigorously?

Reviewer #2: Yes

4. Have the authors made all data underlying the findings in their manuscript fully available?

Reviewer #2: Yes

5. Is the manuscript presented in an intelligible fashion and written in standard English?

Reviewer #2: Yes

Reviewer #2: Thank you for your careful review of the manuscript and for your detailed responses to each of the points raised. Looking at the manuscript again, I feel that it has been substantially improved in terms of clarity, especially the methodological part of the study. In particular, the experimental design as a whole now appears to be much better. Overall, therefore, the revised manuscript is more balanced and methodologically sound, and the presentation of the results is also more consistent with the limitations of the experimental design. The only revision suggestion I would like to make is to make the discussion sections more fluid in relation to the actual conclusion of the text.

**Do you want your identity to be public for this peer review?** For information about this choice, including consent withdrawal, please see our Privacy Policy

Reviewer #2: No

---

## [Editor Report · Acceptance letter]

PONE-D-25-49948R1

PLOS One

Dear Dr. Chang,

I'm pleased to inform you that your manuscript has been deemed suitable for publication in PLOS One. Congratulations! Your manuscript is now being handed over to our production team.

Kind regards,

on behalf of

Dr. Yahya Al Naggar

Academic Editor

PLOS One